# A 0.5 V Sub-Threshold CMOS Current-Controlled Ring Oscillator for IoT and Implantable Devices

Andrea Ballo [1], Salvatore Pennisi [1,*], Giuseppe Scotti [2] and Chiara Venezia [1]

1   Dipartimento di Ingegneria Elettrica Elettronica e Informatica (DIEEI), University of Catania, 95125 Catania, Italy; andrea.ballo@unict.it (A.B.); chiara.venezia@phd.unict.it (C.V.)
2   Dipartimento di Ingegneria dell'Informazione Elettronica e Telecomunicazioni (DIET), Sapienza University of Rome, 00184 Rome, Italy; giuseppe.scotti@uniroma1.it
*   Correspondence: salvatore.pennisi@unict.it; Tel.: +39-095-7382318

**Abstract:** A current-controlled CMOS ring oscillator topology, which exploits the bulk voltages of the inverter stages as control terminals to tune the oscillation frequency, is proposed and analyzed. The solution can be adopted in sub-1 V applications, as it exploits MOSFETS in the subthreshold regime. Oscillators made up of 3, 5, and 7 stages designed in a standard 28-nm technology and supplied by 0.5 V, were simulated. By exploiting a programmable capacitor array, it allows a very large range of oscillation frequencies to be set, from 1 MHz to about 1 GHz, with a limited current consumption. Considering, for example, the five-stage topology, a nominal oscillation frequency of 516 MHz is obtained with an average power dissipation of about 29 μW. The solution provides a tuneable oscillation frequency, which can be adjusted from 360 to 640 MHz by controlling the bias current with a sensitivity of 0.43 MHz/nA.

**Keywords:** ring oscillator; body biasing; tuning range

## 1. Introduction

The Internet of Things (IoT), wireless sensor networks, and the emergence of other energy-harvested microsystems pose continuous challenges and create ever-growing interest in CMOS ultra-low-power analog and mixed-signal system-on-chip solutions. In this framework, applications such as wearable and implantable medical devices, body sensor networks, etc., often require a controlled oscillator (CO) with a minimum power consumption, small layout area, low phase noise, and adequate frequency tuning range to cope with process and/or temperature variations [1–6]. COs are also fundamental blocks of phase-locked loops (PLLs) to provide the timing basis in clock control, clock generator circuits, RFID tags, and systems that use clock-dependent circuits, such as switching power converters and so on [7–9].

CMOS COs can be categorized in two main families. The first includes LC resonant oscillators and, the second, ring oscillators. LC oscillators are mainly used in applications where both a high-phase noise and quality factor (Q) are required. Due to their spiral inductors' large area and high-power dissipation, they cannot be used in ultra-low-power systems on a chip and where physical dimensions must be limited [10]. As is well known, ring oscillators (ROs) consist of an odd number of cascaded delay elements, usually identical to each other, that form a ring where the last stage output is connected to the first stage input. A further categorization is performed based on the control variable, which is often a voltage or a current.

Controlled ring oscillator topologies exhibit a good frequency tuning range, low power dissipation, low design complexity, occupy a small area, and compatibility with CMOS processes. Moreover, ring oscillators are more power efficient compared to relaxation oscillators, although these can achieve a wider tuning range [11].

To improve the frequency-tuning range and phase-noise margin, several design approaches for low-power COs have been reported in the literature. Among these techniques, we mention the combined current-starving technique (i.e., current-controlled oscillator) with a negative skewed-delay approach to improve the power delay product (i.e., product between dissipated power and single gate delay) [12]. A conventional voltage controlled oscillator (VCO) with a negative resistance, multiple-gated circuit and bypass capacitor to suppress high-order harmonics has been reported [13], whereas a digital control circuit to manage oscillation frequency has also been described [1]. An approach that uses positive feedback in each stage to operate with only two stages, instead of three, decreasing the power consumption can be found [14], whereas a frequency tuning cell that consists of one NMOS and one PMOS to form a transmission gate, used to tune the oscillation frequency by varying the gate voltage, has been presented [15]. Finally a dynamic threshold technique (DTMOS) to reduce the threshold voltage of NMOS transistors in the inverting stage with the aim to achieve fast transition and high operating frequency has been presented [3].

The idea of exploiting the bulk terminal to control the oscillation frequency of a RO and the effect of bulk voltage variations on RO phase noise have been analyzed [16], whereas an adaptive body-bias generator for low voltage CMOS VLSI circuits in which a RO was used to estimate the delay of CMOS gates has also been presented [17].

In this work, we exploit a body-biasing technique, originally utilized in the analog domain [18–20], and recently applied to set the quiescent current of the generic inverter stage [21] to design a low-power low-voltage current-controlled ring oscillator (CCO) in 28-nm bulk CMOS technology.

The proposed approach allows to guarantee a static output voltage equal to half the supply voltage, in spite of the value of the bias current, which is tuned in order to control the oscillation frequency. In this way, since any offset in the input output voltage transfer characteristic is removed by the body bias loop, the inverter stages can be reliably cascaded, thus greatly enhancing the tuning range and robustness to PVT variations of the proposed RO.

The manuscript is organized as follows. Section 2 describes the proposed solution. Section 3 reports accurate small- and large-signal analyses of the proposed oscillator. Section 4 includes some simulation results and, finally, the authors' conclusions are summarized in Section 5.

## 2. The Proposed Solution

Figure 1 shows a circuit schematic of the proposed current-controlled ring oscillator (CCRO). It consists of an $N$-stage ring oscillator in which the single stage is made up of a CMOS inverter where bulk terminals of both transistors ($M_{Pi}$ and $M_{Ni}$, with $i = 1, 2, \ldots N$ and $N$ an odd number greater than 3) are made accessible. An output capacitor, $C_i$ in the red-dashed box, is added at the output of each stage with the aim of setting the nominal oscillation frequency (coarse tuning), and to locally make the single stage insensible to parasitic capacitances, as will be clarified in the next section. The body potentials of both transistors, $V_{BP}$ and $V_{BN}$, are generated from the auxiliary topology depicted in Figure 2, the aim of which is to establish the maximum current flowing in the reference inverter ($M_{PR}$-$M_{NR}$), i.e., when the input is at the logic threshold, $V_{DD}/2$. For this purpose, in quiescent conditions, the input terminal of this reference inverter is set to $V_{DD}/2$ and, thanks to the overall negative feedback implemented by error amplifier $A_2$, such condition is transferred also to the output. Note that also the drain voltage of transistor $M_{PA}$ is kept to $V_{DD}/2$ thanks to $A_1$. This allows us to set the same nominal operating points for both $M_{PR}$ and $M_{PA}$.



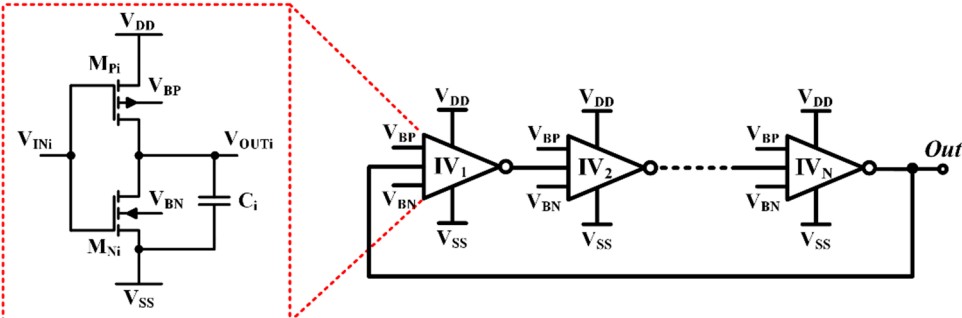

**Figure 1.** Simplified schematic of the proposed current-controlled ring oscillator.

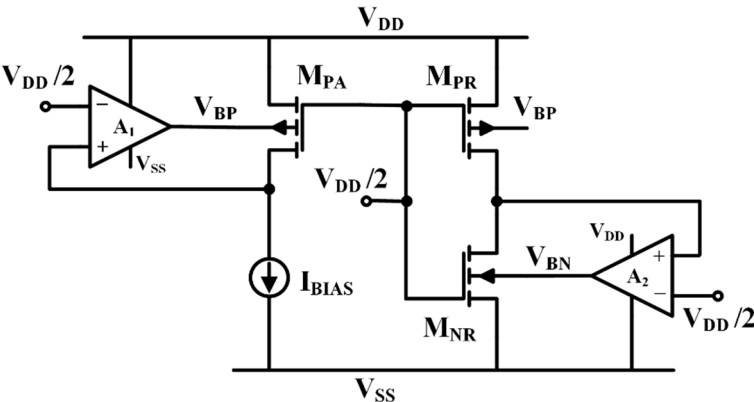

**Figure 2.** Simplified schematic of the biasing section generating $V_{BN}$ and $V_{BP}$ for the RO in Figure 1.

As far as the quiescent current control is concerned, it is implemented through the bulk terminals via voltage $V_{BP}$ for the p-channel transistors, and $V_{BN}$ for the n-channel ones. These voltages are generated by $A_1$ and $A_2$, exploiting a technique proposed in [19] and utilized also in [21].

In brief, starting from the biasing current $I_{BIAS}$, transistor $M_{PA}$ is forced to generate voltage $V_{BP}$, which is also applied to $M_{PR}$. Therefore, current $I_{BIAS}$ in $M_{PA}$ is mirrored by transistor $M_{PR}$ that, as already stated, together with $M_{NR}$, constitutes the reference inverter. Note also that $A_2$ generates the required bulk voltages, $V_{BN}$, for $M_{NR}$ to drive the same current of $M_{PR}$ under the constraints listed in the following:

(a)    assigned aspect ratios $(W/L)_{PA}$, $(W/L)_{PR}$ and $(W/L)_{NR}$;
(b)    $I_{D,PR/NR} = kI_{BIAS}$, where $k = (W/L)_{PR}/(W/L)_{PA}$;
(c)    $V_{SG,PR} = V_{GS,NR} = V_{DD}/2$;
(d)    $V_{SD,PR} = V_{DS,NR} = V_{DD}/2$, assuming ideal input virtual short in $A_1$ and $A_2$.

Of course, aspect ratios of $M_{PR}$ and $M_{NR}$ must be set so that the required bulk voltages are within $V_{DD}$ and ground. Moreover, the mirroring error between the biasing branch and the reference one is reduced using a careful layout style.

It should be noted that the auxiliary amplifiers $A_1$ and $A_2$ should provide a maximum (rail-to-rail) output voltage range, whereas input common mode range is not a concern as input voltage is kept constant to $V_{DD}/2$. Therefore, simple symmetrical OTAs biased in subthreshold, can be effectively used. An example of implementation of this type of amplifier is found in [21,22], and is shown in Figure 3.

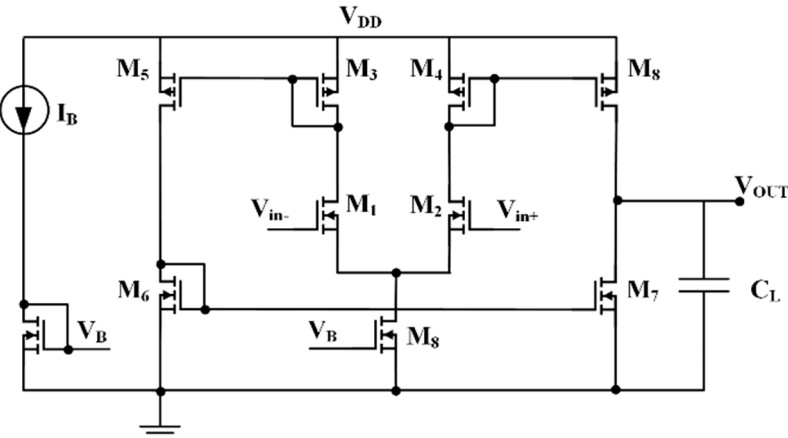

**Figure 3.** Simplified schematic of simple mirror OTA [21] used in this work.

Voltages $V_{BN}$ and $V_{BP}$ are then applied to the inverters forming the ring oscillator in Figure 1, limiting at the desired value the maximum current flowing when the input voltage is equal to the threshold. Indeed, consider transistor $M_{N1}$ of the first inverter stage, the exploded view of which is depicted in the red-dashed box in Figure 1. Let us remember that, in quiescent conditions, $V_{IN1}$ is equal to $V_{DD}/2$. Consequently, $M_{NR}$ and $M_{N1}$ have respectively the same source, gate, and bulk voltage and hence the drain current of $M_{N1}$ is related to that of $M_{NR}$ in a mirror-like condition:

$$I_{D,N1} = \frac{(W/L)_{N1}}{(W/L)_{NR}} I_{BIAS} \tag{1}$$

where equality is accurately verified because the source-drain voltage of $M_{N1}$ is also equal to $V_{DD}/2$. Similar considerations hold for all the transistors in the ring oscillator, in practice, all p-channel and n-channel devices have their current linked to $I_{BIAS}$ via the current-mirror relations

$$I_{D,Pi} = \frac{(W/L)_{Pi}}{(W/L)_{PR}} I_{BIAS} \tag{2}$$

$$I_{D,Ni} = \frac{(W/L)_{Ni}}{(W/L)_{NR}} I_{BIAS} \tag{3}$$

where $(W/L)_{Pi}$ and $(W/L)_{Ni}$, with $i = 1, 2, \ldots N$, are, respectively, the aspect ratios of the generic p-channel and n-channel MOSFET in the ring oscillator.

### 3. Small- and Large-Signal Analysis of the Proposed Ring Oscillator

In order to design a conventional ring oscillator, analytical extraction of design equations is carried out by using two main approaches.

The first type of analysis considers small-signal equivalent model of the sub-blocks and Barkhausen stability criterion. In this approach the single gate is seen as working in an operating point (biasing or linearity conditions) and the whole system is analysed in the frequency domain. For this reason, hereinafter we will refer to this approach as an *analog* or *small-signal approach*. As an example, let us consider the conventional CMOS inverter in Figure 4 and its equivalent small-signal circuit.

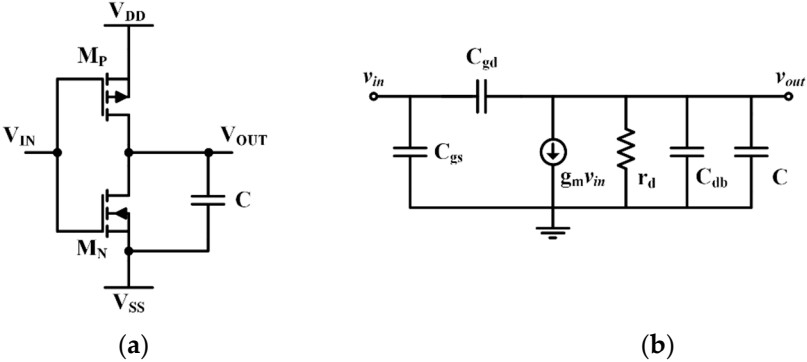

**Figure 4.** Conventional CMOS inverter (**a**) and its equivalent small-signal circuit (**b**).

When working around an operating point, the inverter behaves as the linear network reported on the right side of Figure 4, the parameters of which are expressed below for the MOS transistors operated in the sub-threshold region.

$$g_m = g_{m,p} + g_{m,n} \simeq 2\frac{I_D}{nV_T} \tag{4}$$

$$r_d = r_{d,n} \parallel r_{d,p} = \frac{nV_T}{2\lambda_{DS}I_D} \tag{5}$$

$$C_{gs} = C_{gs,p} + C_{gs,n} \simeq \frac{2}{3}C_{OX}(W_pL_p + W_nL_n) + C_{OX}(W_pL_{ov} + W_nL_{ov}) \tag{6}$$

$$C_{gd} = C_{gd,p} + C_{gd,n} \approx C_{OX}(W_pL_{ov} + W_nL_{ov}) \tag{7}$$

$$C_{db} = C_{db,p} + C_{db,n} \approx 2C_{Jp/n}\Big|_{V_{DD}/2}\left[1 - \frac{1}{m_j}\frac{V_{DD}/2}{V_{bi}}\left(1 - \frac{V_{DD}/2}{V_{bi}}\right)\right] \tag{8}$$

where parameter $n$ is the sub-threshold slope, $V_T = kT/q$ is the thermal voltage, with $k$ the Boltzmann constant, $T$ is the absolute temperature and $C_{OX}$ is the oxide capacitance for unit of area. In addition, $\lambda_{DS}$ is the channel modulation coefficient, $L_{ov}$ is the length of the overlap portion, $C_{Jp/n}$ is the capacitance of the S/D junctions (evaluated at the voltage $V_{DD}/2$ in (4e)), $m_j$ is the grading coefficient and $V_{bi}$ is the built-in voltage.

The product between the transconductance $g_m$ (4) and the output resistance $r_d$ (5) yields a constant value, independent of the biasing current $I_D$ and equal to the maximum of the $g_m/I_D$ curves [23]. In such case, only the channel modulation coefficient, $\lambda_{DS}/nV_T$, can be changed by sizing the transistors, in order to (slightly) change the inverter intrinsic gain (i.e., $g_mr_d$). Note that the drain-induced barrier lowering (DIBL) effect is included in the channel modulation coefficient through the parameter $\lambda_{DS}$. Parasitic capacitance contribution accounts for three capacitances expressed in (6)–(8).

The gate-to-source equivalent capacitance is proportional to $C_{OX}$ and is constituted by a first term that depends on the MOSFET active areas and by the operating condition (assumed with MOSFETs in saturation) and a second term that depends on the overlap capacitance. A similar contribution forms the gate-to-drain equivalent capacitance, $C_{gd}$, expressed in (7). The drain-to-bulk capacitance, unlike the previous two, is a non-linear capacitance which depends on S/D diffused areas (included in $C_{Jp/n}$) and the applied voltage, i.e., the drain-to-bulk voltage. Referring to Figure 4a, both are evaluated in the quiescent point, i.e., at $V_{DD}/2$, and, from Figure 4b, the output node results to be loaded by the sum of the capacitances (8) and the additional one, $C$.

The above derivation, when applied to the proposed circuit, yields the same equations except for (4e) that becomes:

$$C_{db} = C_{db,p} + C_{db,n} \approx 2\,C_{Jp/n}\Big|_{V_{DD}/2-V_B} \left[ 1 - \frac{1}{m_j} \frac{V_{DD}/2 - V_B}{V_{bi}} \left( 1 - \frac{V_{DD}/2 - V_B}{V_{bi}} \right) \right] \quad (9)$$

However, the effect of $C_{db}$ on the oscillation frequency can be neglected if an additional capacitance, $C$, sufficiently large, is connected in parallel. Analysis of the complete ring oscillator leads to closed-loop gain and phase shift which satisfy Barkhausen's criteria for the common pulsation, $\omega_p$, since the output node electrically coincides with the input one, therefore $|H(j\omega_p)| = 1$, and the a total phase shift of 180° is constantly achieved for an odd number of stages $N$. The result of these concurrent conditions ensures oscillation whose frequency is expressed by:

$$f_{OSC} = \frac{\omega_P}{2\pi} = \frac{1}{2\pi r_d c_{tot}} \tan\left(\frac{\pi}{N}\right) \quad (10)$$

Here $c_{tot}$ gathers all the capacitive contributions (6), (7) doubled for Miller's effect, (9) and $C$. It can be noted that, being the output small-signal resistance inversely proportional to the biasing current, $I_D$, a proportional control of the oscillation frequency can be operated by varying the current itself. Various works presented in literature demonstrated that such kind of analysis is inaccurate when the number of stages exceeds 3, hence (10) is rarely used to design a ring oscillator.

On the other hand, the second approach consider the oscillator as the cascade of an odd-number of digital inverting gates where the output of the last gate is fed-back to the input of the first one. In this framework, the single inverter is characterized by its propagation delay, $\tau_{PD}$, and the frequency of the generated signal follows the expression:

$$f_{OSC} = \frac{1}{2N\tau_{PD}} \quad (11)$$

where the factor 2 derives from the fact that each single voltage node switches $N$-times $\tau_{PD}$, where $N$ is the number of inverters involved. For a digital gate, the propagation delay is defined as the time required to settle the output node to the middle of its dynamic range as referred to the instant of input changing. Henceforth, we call this approach *digital* or *large-signal approach*. While the simple relation in (11) and its scalability assuming general gate implementation are the strengths of this approach, evaluating $\tau_{PD}$ could require a great deal of effort. Therefore, designers often adopt a trial-and-error approach.

To better understand the relation between small- and large-signal behavior, propagation delay of the proposed cell should be evaluated. Figure 5 shows the working principle of the inverting gate in response to an input rail-to-rail signal and its static behavior as well. Assuming the inverter symmetrical and working in sub-threshold, which means to size transistors aspect ratios meeting the relationship

$$\left(\frac{W}{L}\right)_P \Big/ \left(\frac{W}{L}\right)_N = \frac{I_{ST0,N}}{I_{ST0,P}} e^{\frac{|V_{TH,P}| - V_{TH,N}}{nV_T}} \quad (12)$$

where both $I_{ST0,N}$ ($I_{ST0,P}$), defined as the potential sub-threshold current of the NMOS (PMOS) if the threshold voltage are nullified, and $n$ are technology-dependent parameters, and $V_{TH,N}$ ($V_{TH,P}$) are the threshold voltage of the involved transistors. In (12), the tailing effect of drain-to-source voltages is neglected because we assume that transistors are biased in saturation, i.e., $V_{DD}/2 > V_T$. Moreover, $V_{TH,N}$ ($V_{TH,P}$) implicitly depends on $V_{BS,N}$ ($V_{SB,P}$) through the body effect, as well as on $V_{DS,N}$ ($V_{SD,P}$) through the DIBL effect. Their contributions are taken into account by expressing $V_{TH,N} = V_{TH0,N} - \lambda_{BS,N} V_{BS,N} - \lambda_{DS,N} V_{DS,N}$ ($|V_{TH,P}| = |V_{TH0,P}| - \lambda_{BS,P} V_{SB,P} - \lambda_{SD,P} V_{SD,P}$) where $V_{TH0,N}$ ($V_{TH0,P}$) and $\lambda_{BS,N}$ ($\lambda_{BS,P}$) are two technology parameters, while $\lambda_{DS,N}$ ($\lambda_{SD,P}$) coincides with that used in



(5) [24]. It should be noted that, if (12) is fulfilled, the two transistors are equally strong, which means that for the same gate to source voltage they conduct the same current. Under the aforementioned considerations, a good approximation (typical error < 10%) for the propagation delay is given by [25]:

$$\tau_{PD} = \frac{(V_{DD}/2)C_{TOT}}{I_{ST}|_{V_{DD}=0}\,e^{\frac{V_{DD}(1+\lambda_{DS,N}/2+\lambda_{SD,P}/2)+\lambda_{BS,N}V_{BN}+\lambda_{SB,P}(V_{DD}-V_{BP})}{nV_T}}} = \frac{(V_{DD}/2)C_{TOT}}{I_D\,e^{\frac{V_{DD}/2}{nV_T}}} \tag{13}$$

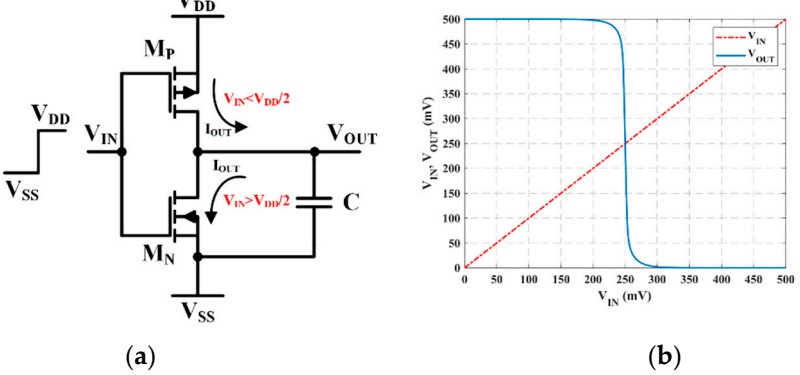

**Figure 5.** Conventional CMOS inverter (**a**) and its static transfer behavior (**b**).

In the first expression of (13), almost all the technology-dependent characteristics and transistor sizes are gathered in $I_{ST}|_{VDD=0}$ in order to be enucleated from the circuital parameters like voltages $V_{DD}$, $V_{BN}$, and $V_{BP}$. Moreover, the total large-signal capacitance seen at the output node, $C_{TOT}$, can be assumed to be equal to the small-signal one reported in (10). Finally, (13) has be re-written in the last simple form to highlight the biasing current, $I_D$.

Replacing (13) in (11), the oscillation frequency is expressed as:

$$f_{OSC} = \frac{I_D}{2N(V_{DD}/2)C_{TOT}}e^{\frac{V_{DD}/2}{nV_T}} \tag{14}$$

As compared with the small-signal counterpart, (14), like (10), shows a linear dependence with the bias current, therefore confirming the possibility to modulate the oscillation frequency of the RO by using the biasing circuit in Figure 2. It should be noted that (14) and (10) give similar information also when the last one loses accuracy. In fact, for $N > 3$ the tangent function can be expanded in Taylor's series, $Tan(\pi/N) \approx \pi/N$ being $\pi/N << 1$. This approximation leads to have:

$$f_{OSC} = \frac{\omega_P}{2\pi} \approx \frac{1}{2Nr_dc_{tot}} = \frac{I_D}{2Nc_{tot}}\left(\frac{2\lambda_{DS}}{nV_T}\right) \tag{15}$$

which differs from (14) only for factor $(\lambda_{DS}/nV_T)$ that replaces $(e^{\frac{V_{DD}/2}{nV_T}}/V_{DD})$. Thus, it can be claimed that small-signal and large-signal analyses yield results that are similar to those obtained for a conventional topology, such as the current-starved RO [26].

In conclusion, three important metrics for a controlled oscillator are evaluated. Starting from (14), the first is the frequency-to-current first-order slope defined as the derivative function of the frequency versus the control current:

$$\frac{\partial f_{OSC}}{\partial I_D} = \frac{e^{\frac{V_{DD}/2}{nV_T}}}{2N(V_{DD}/2)C_{TOT}} \tag{16}$$

The second one is the total power consumption, made up of a static and a dynamic contribution. While the static part is due only to the leakage current, which coincides

with the quiescent one in our case and, as it will be seen it is negligible; the dynamic part represents the major contribution to the power consumption. Consequently, the dynamic power dissipation $P_D$ of a $N$-stage ring oscillator is given by:

$$P_D = N C_{tot} f_{OSC} (V_{DD})^2 \tag{17}$$

Finally, in a conventional CMOS oscillator, the amount of the phase noise, $L\{\Delta f\}$, (see expression (15) of [27]) is given by the flicker noise and its normalized single-sideband spectral density as given in the following equation

$$L\{\Delta f\} = 10 \log \left[ \frac{2FkT}{P_{sign}} \left( \frac{f_{OSC}}{2Q\Delta f} \right)^2 \right] \tag{18}$$

where $\Delta f$ is the offset frequency from the nominal one $f_{OSC}$, $Q$ is the quality factor and $F$ is an empirical fitting parameter that takes the increased noise in $\Delta f$ into account. The $Q$ factor is typically used in the design of high-order oscillators like $LC$-type and is defined as the ratio of the energy stored in the oscillating resonator to the energy dissipated per cycle by damping processes. Finally, $P_{sign}$ in (18) is the power of generated signal. Unfortunately, as in the conventional ring oscillator, the quality factor is poor since the energy stored in the node capacitances is reset(discharged) every cycle [27], resulting in a higher phase noise.

Finally, the trade-off between phase noise, power consumption and carrier frequency can be evaluated by using the following figure of merit (*FoM*):

$$FoM = L\{\Delta f\} + 10 \log \left( P_{(mW)} \right) - 20 \log \left( \frac{f_{OSC}}{\Delta f} \right) \tag{19}$$

where $P_{(mW)}$ is the power consumption expressed in mW, thus normalized to 1 mW.

## 4. Validation Results

The proposed solution in the version of 3-, 5- and 7-stage CROs was designed in a 28-nm triple-well CMOS technology provided by TSMC and simulated at the schematic level. To set symmetrical behavior of the inverter, of the control bulk voltages ranges and body effect coefficients, MOS transistors with different thresholds were exploited. Specifically, HVT (high threshold) n-channel with 515-mV $V_{TH}$ and SVT (standard threshold) p-channel devices with −460-mV $V_{TH}$, were adopted. A single power supply of 0.5 V was set and $I_{BIAS}$ was 320 nA. Reference operating temperature was in the range from 0 °C to 60 °C, suitable for implanted and wearable circuits. Transistor dimensions, together with other component values, are summarized in Table 1.

**Table 1.** Design parameters used in simulations.

| Parameter | Value |
|---|---|
| $V_{DD}$ | 0.5 V |
| $I_{BIAS}$ | 320 [a] nA |
| $(W/L)_{PA}$, $(W/L)_{PR}$, $(W/L)_{Pi}$ | 8.28/0.18 μm/μm |
| $(W/L)_{NR}$, $(W/L)_{Ni}$ | 5.4/0.18 μm/μm |
| $A_1$, $A_2$ | 30 dB |
| $GBW_{A1}$, $GBW_{A2}$ | 10 kHz |
| $C_i$ | 10 fF |

[a]: This value will be changed to 350 nA after corner analysis.

All p-channel (n-channel) MOSFETS are equal to the reference device 8.28/0.18 (5.4/0.18) μm/μm, where channel length was slightly increased as respect to the minimum one (100 nm) to counteract the short-channel effect. With these design choices the

mirroring coefficient, $k_i$, and the ratios of the transistors' form factors in (1)–(3) are all reduced to the unity. As a consequence of the transistor's dimension, the nominal quiescent current in each branch, which coincides with its short-circuit current, of 320 nA, resulting in a total nominal quiescent current of $N$-times 320 nA. Coarse tuning capacitor $C_i$ was set to 10 fF for all stages. The DC gain of the auxiliary amplifiers, $A_1$ and $A_2$, with transistors in subthreshold, was around 30 dB and the gain-bandwidth product was 10 kHz, while consuming only 50 nA.

The robustness of the quiescent conditions was validated at first. The nominal bulk voltages, $V_{BP}$ and $V_{BN}$, generated by the circuit in Figure 2 were 251 mV and 249 mV, respectively. The simulated quiescent current in the main ring oscillator in Figure 1 was 961, 1602 and 2243 nA on average, with a standard deviation of 48.5, 78.3, and 107 nA, respectively, for 3-, 5-, and 7-stage topology after running 1000 Monte Carlo iterations. The difference with respect to the expected values is mainly due to the low DC gains of the auxiliary amplifiers, which cause a closed-loop gain error.

Figure 6a shows the body voltages of the transistors involved in the reference inverting gate for a sweeping of the biasing current, $I_{BIAS}$, in the range 120 nA–820 nA. The voltages fall within the supply rails and, in particular, it is easy to observe that their behaviors are symmetrical, confirming a good sizing of the block and the possibility to exploit the full dynamic range of the control voltages. Currents entering in the body terminals have been also evaluated and reported in Figure 6b to highlight that body junctions are never fully turned on during the control operation. In fact, the values of body currents in the worst case (PMOS), reach around 10 nA, corresponding to less than 2% of the biasing one.

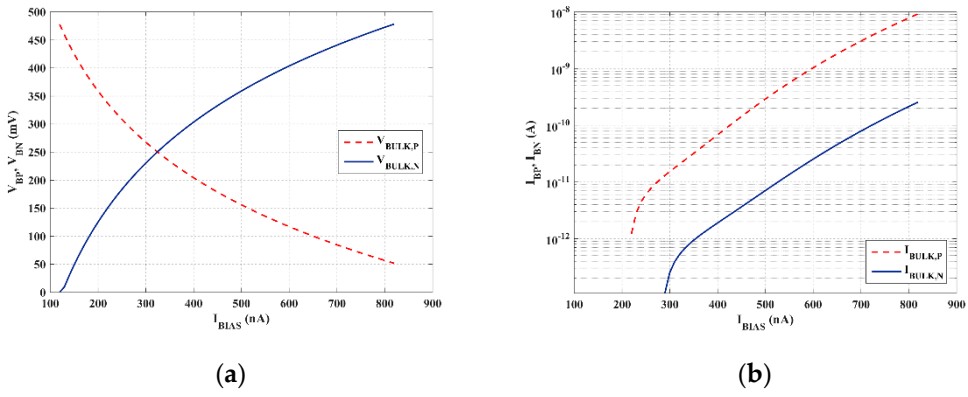

**Figure 6.** Body voltages (**a**) and currents (**b**) in the interested current biasing range at $T = 30\ °C$.

Figure 7a depicts the current flowing in the reference inverting gate when its input is varied from 0 to $V_{DD}$ (500 mV). As expected, the maximum is achieved for $V_{DD}/2$ and accurately follows $I_{BIAS}$ as a validation of the effectiveness of the exploited biasing strategy and the linearity of the relation between the two quantities as well.

Figure 8 illustrates the output signal of the 5-stage CRO with $C_i = 10$ fF for three values of biasing current, 120, 320, and 820 nA, representing the minimum, nominal and maximum value, respectively.

Figure 9 shows the oscillation frequency as a function of biasing current ($C_i = 10$ fF). An oscillation range from 360 MHz to 640 MHz is found with tuning sensitivity, i.e., the ratio between $(f_{MAX} - f_{MIN})/(I_{BIAS,MAX} - I_{BIAS,MIN})$, about equal to 0.43 MHz/nA. Compared with the predicted behavior (linear relationships resulting from (14) and (15)), the obtained one shows a logarithmic relationship with $I_{BIAS}$. This is confirmed by the inset plot in the same figure, the x-axis of which is logarithmic and slightly extended to cover an entire decade. Such changing in the behavior may be due to the partial operation in moderate inversion region, where subthreshold equations lose accuracy.

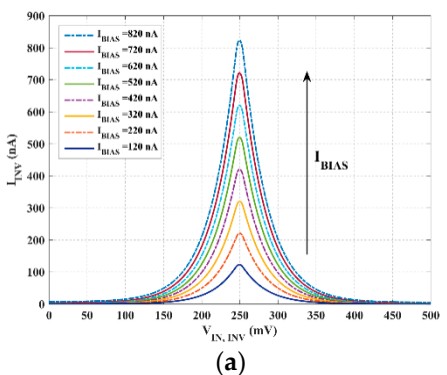
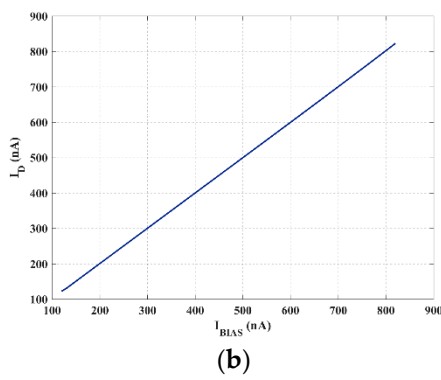

(**a**) (**b**)

**Figure 7.** Static currents flowing in the reference inverter vs. input voltage for different $I_{BIAS}$ (**a**), and static currents maxima vs. $I_{BIAS}$ ($T = 30$ °C) (**b**).

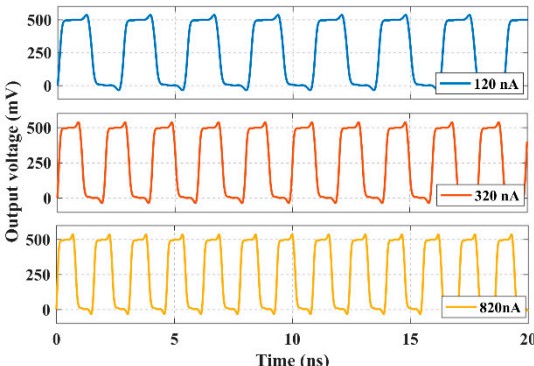

**Figure 8.** Output signal of 5-stage CRO for three different values of biasing current at $T = 30$ °C.

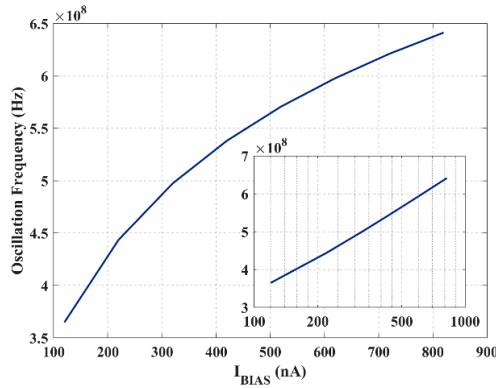

**Figure 9.** Oscillation frequency of the 5-stage CRO as a function of the biasing current at $T = 30$ °C.

Figure 10a,b shows oscillation frequency with number of stages $N$ equal to 3, 5, and 7 and for different coarse tuning capacitances, $C_i$, in the considered current biasing range (Figure 10a) and for a fixed $I_{BIAS} = 320$ nA (Figure 10b) at $T = 30$ °C. It is apparent that frequency varies with the bias current, independently of the number of stages and coarse tuning capacitance. Constant spacing between two adjacent curves shows that the number of stages $N$ acts as a scaling constant factor in the expression of the frequency, as predicted by (14) or, equivalently, (15). Moreover, Figure 10b highlights that the coarse tuning capacitance is comparable, in the range between 10 fF and 100 fF, with the parasitic inverter capacitances, being the oscillation frequency to capacitance relation compressed in this range.



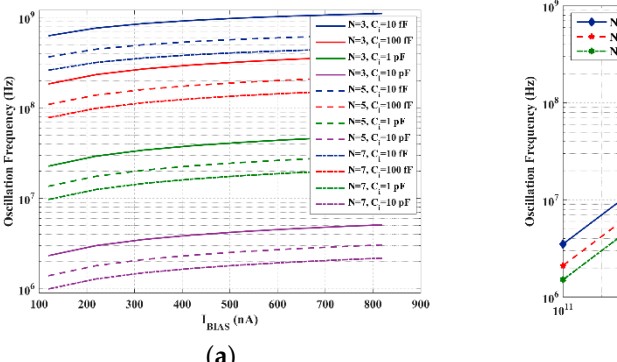 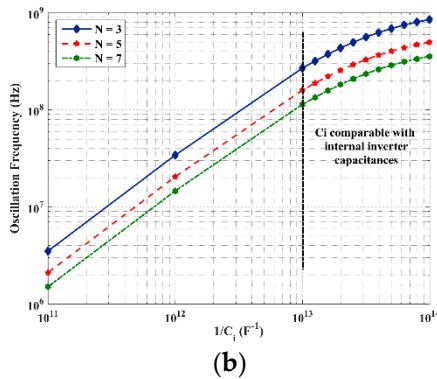

(a)            (b)

**Figure 10.** Oscillation frequencies for different coarse tuning capacitance values in the interested current biasing range (**a**) and for a fixed $I_{BIAS}$ = 320 nA (**b**) at $T$ = 30 °C.

Figures 9 and 10 also show that the proposed solution may be used in an automatic design procedure, which, starting from the oscillation frequency specification and the nominal bias current, allows to determine the number of inverter stages (which can be taken from a standard-cell library providing access to the bulk terminals) and the coarse tuning capacitances (which can be taken from a capacitor array). The bias current is then used to perform oscillation frequency tuning to counteract process and temperature variation effects.

At this purpose, corner analyses were carried out for the 5-stage topology as an illustrative example. Figure 11 shows the two bulk-source voltages $V_{BN}$ and $V_{BP}$ as a function of $I_{BIAS}$, at 30 °C. It is apparent that to ensure correct operation (i.e., to maintain bulk voltages within the supply limits) biasing current range must be limited from 240 nA to 460 nA.

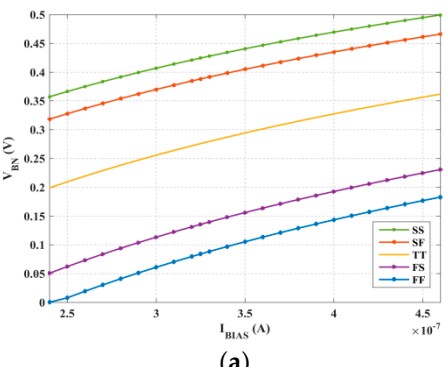 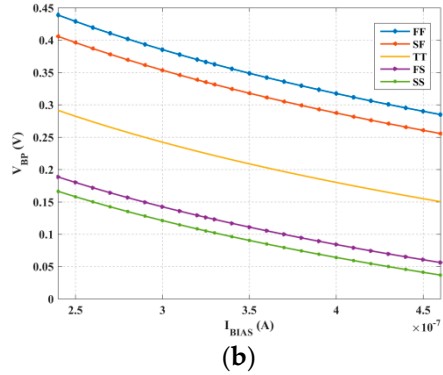

(a)            (b)

**Figure 11.** Bulk−source voltage of NMOS (**a**) and PMOS (**b**) transistors vs. biasing current over the 5 basic process corners.

Specifically, NMOS transistors experience the highest process variations. In fact, the upside limit of the current range under slow NMOS corners (SS or SF) must be limited to 460 nA. Vice versa, under fast NMOS corners (FF or FS), $I_{BIAS}$ current must be larger than 240 nA.

Figure 12 shows the simulated oscillation frequency of the 5-stage ring oscillator in this range of $I_{BIAS}$, for the five basic corners (at 30 °C). It can be noted that the tuning frequency range is independent of the process corners. Indeed, the tuning sensitivity is constant regardless the corner and is still approximately equal to 0.43 MHz/nA. The maximum percentage variation between the nominal oscillation frequency and that affected by corners is about 20%.

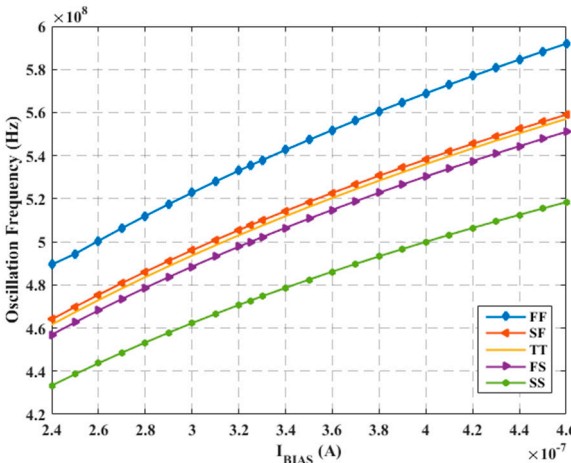

**Figure 12.** Oscillation frequency vs. biasing current over the 5 basic process corners.

Phase noise versus offset frequency ($\Delta$f) for the basic corners is illustrated in Figure 13, which clearly shows the close overlap of the five curves, indicating that also the phase noise is process independent.

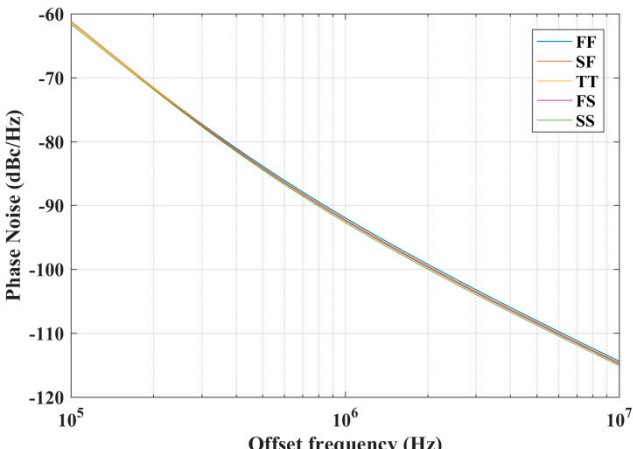

**Figure 13.** Phase noise vs. offset frequency over the 5 basic process corners.

Phase noise was also simulated in the 0 °C to 60 °C temperature range and across the five corners. The minimum value of the simulated phase noise is −91.86 dBc/Hz when $T = 0$ °C in the FF corner, while the maximum value of the simulated phase noise is −93.72 dBc/Hz when $T = 0$ °C in the SS corner (both values are evaluated at the nominal bias current $I_{BIAS}$ equal to 350 nA).

Tables 2–4 summarize the corner analysis results of the main parameters of the simulated 5-stage current-controlled ring oscillator (nominal $I_{BIAS}$ equal to 350 nA and at 30 °C) for three values of the supply voltage $V_{DD}$.

**Table 2.** Corner analysis of the 5-stage current-controlled ring oscillator performed at 30 °C and at $V_{DD} = 475$ mV.

| Corner | TT | FF | FS | SF | SS |
|---|---|---|---|---|---|
| Oscillation frequency (MHz) | 451.2 | 481.6 | 446.2 | 446.7 | 397.3 |
| Tuning range (MHz) | 87.09 | 96.38 | 50.69 | 59.24 | 24.12 |
| Phase noise @1 MHz (dBc/Hz) | −92.77 | −92.39 | −92.90 | −92.90 | −93.75 |
| Average power consumption (µW) | 23.40 | 24.41 | 23.31 | 23.14 | 20.94 |

**Table 3.** Corner analysis of the 5-stage current-controlled ring oscillator performed at 30 °C and at $V_{DD}$ = 500 mV.

| Corner | TT | FF | FS | SF | SS |
|---|---|---|---|---|---|
| Oscillation frequency (MHz) | 516.2 | 547.3 | 410.7 | 418.5 | 482.5 |
| Tuning range (MHz) | 95.44 | 102.50 | 94.11 | 94.94 | 85.08 |
| Phase noise @1 MHz (dBc/Hz) | −92.47 | −92.13 | −92.58 | −92.40 | −92.87 |
| Average power consumption (μW) | 28.6 | 29.8 | 28.4 | 28.8 | 27.5 |

**Table 4.** Corner analysis of the 5-stage current-controlled ring oscillator performed at 30 °C and at $V_{DD}$ = 525 mV.

| Corner | TT | FF | FS | SF | SS |
|---|---|---|---|---|---|
| Oscillation frequency (MHz) | 583.3 | 619.7 | 577.6 | 586.5 | 549.6 |
| Tuning range (MHz) | 102.47 | 59.27 | 81.37 | 100.50 | 93.52 |
| Phase noise @1 MHz (dBc/Hz) | −92.14 | −91.98 | −92.25 | −92.06 | −92.48 |
| Average power consumption (μW) | 36.16 | 37.93 | 35.85 | 36.29 | 34.61 |

Due to process variations, oscillation frequency varies of about 20%. However, tuning range variations across the five corners are limited to 8% and both phase noise and average power consumption variations are 5%. Regarding power consumption, the typical (TT) value is accurately predicted by (14), where $C_{tot}$ can be estimated to be 88.6 fF, including the additional load capacitance of 10 fF. This value agrees with the results shown in Figure 10b.

Mismatch Monte Carlo simulations of the oscillation frequency in typical conditions ($V_{DD}$ = 0.5 V, $I_{BIAS}$ = 350 nA and $T$ = 27 °C) are reported in Figure 14, showing the limited impact of mismatches on the oscillation frequency of the proposed CRO.

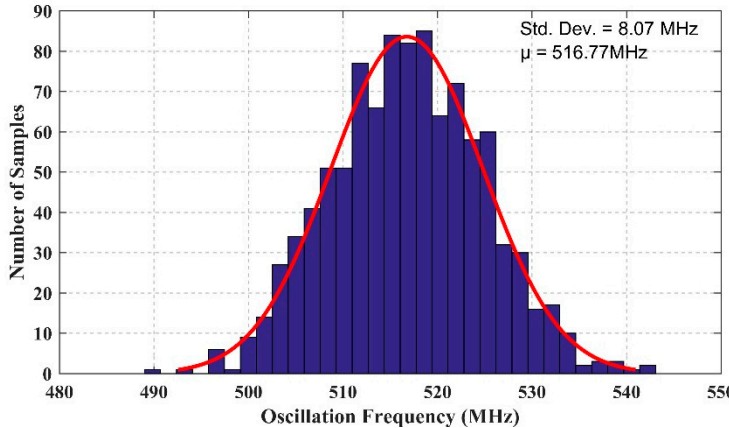

**Figure 14.** Mismatch Monte Carlo simulations of the oscillation frequency in typical conditions ($VDD$ = 0.5 V, $I_{BIAS}$ = 350 nA and $T$ = 27 °C).

As already mentioned, the tuning capabilities of the proposed CRO can be exploited to compensate for the effects of temperature variations (in a non-exclusive alternative to process variations). To give an example, Figure 15 shows some isofrequency curves (at 557 MHz, 538 MHz, 516 MHz, 491 MHz, and 462 MHz) in $I_{BIAS}$ vs. temperature plot. Each point of the curves establishes the current $I_{BIAS}$ needed to set the target frequency between around 450 MHz and 557 MHz in the operating range from 0 °C to 60 °C.

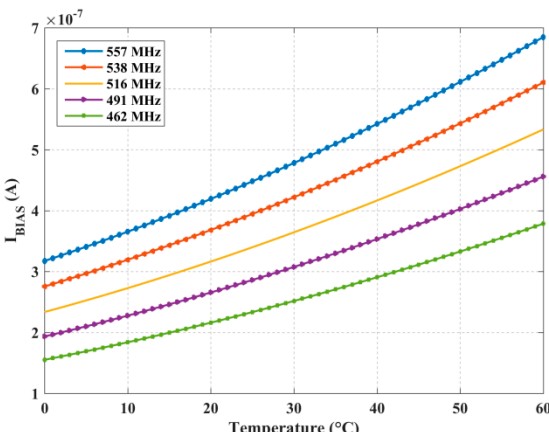

**Figure 15.** Isofrequency curves in the $I_{BIAS}$ vs. temperature plot (TT).

The tuning capabilities of the proposed CRO can be exploited also to compensate the effects of supply voltage ($V_{DD}$) variations. To give an example, Figure 16 shows some isofrequency curves (at 557 MHz, 538 MHz, 516 MHz, 491 MHz, and 462 MHz) in the $I_{BIAS}$ vs. $V_{DD}$ plot. Each point of the curves establishes the current $I_{BIAS}$ needed to set the target frequency between around 450 MHz and 557 MHz in the operating range from 475 mV to 525 mV.

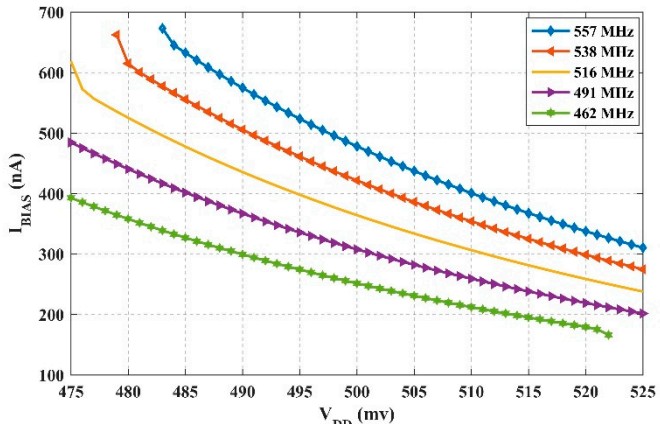

**Figure 16.** Isofrequency curves in the $I_{BIAS}$ vs. $V_{DD}$, (TT).

The layout of the five stage CRO is reported in Figure 17, showing an area footprint of 12.4 μm × 7.5 μm. To better assess the reliability of the above results, post layout simulations have been carried. The main effect of the layout resulted in a parasitic capacitance of 6.5 fF at the output node of the CRO. Once reduced the explicit coarse tuning capacitance of an amount equal to the parasitic capacitance, post layout simulations resulted to be in very good agreement with schematic level simulations.

Table 5 compares some recent controlled oscillator topologies presented in the literature with the proposed one, which provides the best performance in terms of phase noise and power consumption, determining the best FoM as defined by (19).

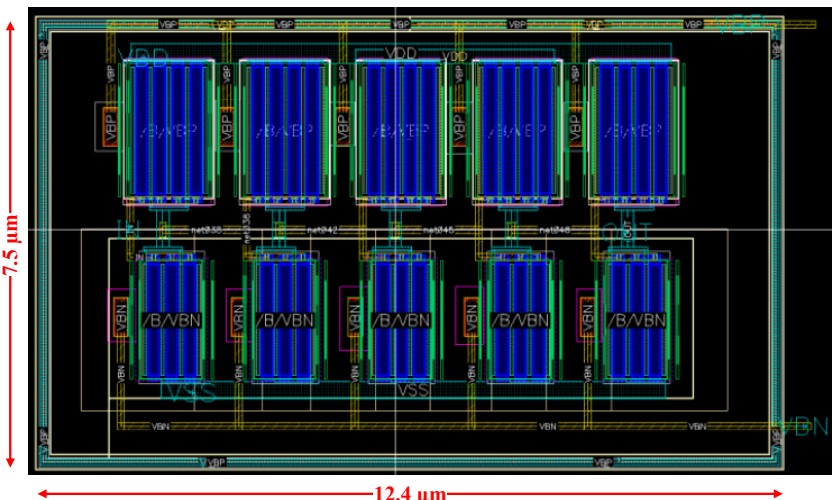

**Figure 17.** Layout of the five stage CRO.

**Table 5.** Comparison with the state-of-the-art.

| Reference | [28] | [29] | [30] | This Work [b] |
|---|---|---|---|---|
| Tech. (nm) | 180 | 65 | 65 | 28 |
| $V_{DD}$ (mV) | 500 | 600 | 700 | 500 |
| N stages | 3 | 3 | 4 | 5 |
| Type of control | Voltage | Voltage | Voltage | Current |
| Osc. frequency (MHz) | 82–370 | 250–800 | 880–1360 | 360–640 |
| Phase Noise (dBc/Hz)@1 MHz | −82 | −86.38 | −90 | −92.47 |
| Power consumption (μW) | 60 | 146.2 | 360 | 28.6 |
| FoM [a] (dBc/Hz) | −145.6 | −153.2 | −153.6 | −164 |

[a]: see (16); [b]: simulations.

## 5. Conclusions

A novel approach to tune the delay of the basic inverter cell of CROs has been presented in this paper. The approach allowed to accurately set the maximum current of all the inverters in the CRO through a body bias loop and to tune the oscillation frequency by controlling the value of a reference current. Small-signal and large-signal analysis of the proposed CRO topology have been carried out to provide insight into circuit behavior and to provide useful design equations.

Current controlled ring oscillators made up of 3, 5, and 7 stages have been designed referring to a commercial 28-nm technology and with a supply voltage of 0.5 V.

Simulation results demonstrated that the proposed approach allows to optimize the tradeoff between tuning range, phase noise and power consumption, as demonstrated by the value of the FoM which outperforms all the similar designs in the literature. Extensive parametric and corner simulations have demonstrated a good robustness of the proposed CROs to PVT variations despite the adoption of a very short channel process node.

**Author Contributions:** Conceptualization: S.P. and G.S.; data curation: A.B. and C.V.; original draft preparation: A.B. and C.V.; writing—review and editing: all authors; supervision: S.P. and G.S. All authors have read and agreed to the published version of the manuscript.

**Funding:** This research received no external funding.

**Institutional Review Board Statement:** Not applicable.

**Informed Consent Statement:** Not applicable.

**Data Availability Statement:** The data presented in this study are available in article.

**Conflicts of Interest:** The authors declare no conflict of interest.

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
