# Peer review of "A 0.5 V Sub-Threshold CMOS Current-Controlled Ring Oscillator for IoT and Implantable Devices"

_jlpea, doi:10.3390/jlpea12010016_

Round 1

Reviewer 1 Report

The authors presented a paper entitled "0.5-V Sub-Threshold CMOS Current-Controlled Ring Oscillator for IoT and implantable devices." The initiative to implement the RO with 0.5V is justifiable for IoT devices. Yet, the body biasing technique is not a very novel one. It has been proposed for different purposes (Opamp, switch, etc). Also, only simluation results are presented. Here are my detailed comments:

  1. There are already several papers describing the body-biasing technique to implement RO, such as M.H. Kazemeini, M.J. Deen, and S. Naseh ISCAS 2003 and A. SRIVASTAVA and C. ZHANG International Journal of Distributed Sensor Networks 2008. Hence, I don't see there is much novelty in this article.
  2. More literature about the design of <0.5V oscillators should be included in reference to support this work. 
  3. Why is the reference VDD/2 used in Fig. 2? How did the authors guarantee that this reference will establish the maximum current flowing in the reference inverter? 
  4. Can the authors comment on the operation of the opamp with 0.5V supply? How can you guarantee that all transistors are in the saturation region? 
  5. The output frequency deviation of the oscillator with the presence of process/mismatch should be elaborated. 
  6. The line sensitivity of the oscillator should be discussed. With the 0.5V supply voltage, I suspect that the oscillator is very sensitive to voltage variations. 
  7. How did the oscillating frequency behave amid process variations? 

Author Response

Comments and Suggestions for Authors

The authors presented a paper entitled "0.5-V Sub-Threshold CMOS Current-Controlled Ring Oscillator for IoT and implantable devices." The initiative to implement the RO with 0.5V is justifiable for IoT devices. Yet, the body biasing technique is not a very novel one. It has been proposed for different purposes (Opamp, switch, etc). Also, only simulation results are presented. Here are my detailed comments:

  1. There are already several papers describing the body-biasing technique to implement RO, such as M.H. Kazemeini, M.J. Deen, and S. Naseh ISCAS 2003 and A. SRIVASTAVA and C. ZHANG International Journal of Distributed Sensor Networks 2008. Hence, I don't see there is much novelty in this article.

REPLY: We thank the reviewer for this comment. We have added the suggested references in the revised manuscript. We have also briefly discussed the differences between the proposed approach and previous work in the last part of the introduction of the revised manuscript. Specifically, the proposed approach allows to guarantee a static output voltage equal to VDD/2 in spite of the value of the bias current which is tuned in order to control the oscillation frequency. In this way, since any offset in the input output voltage transfer characteristic is removed by the body bias loop, the inverter stages can be reliably cascaded, thus greatly enhancing the tuning range and robustness to PVT variations of the proposed RO.

  1. More literature about the design of <0.5V oscillators should be included in reference to support this work. 

REPLY: We thank the reviewer for this comment. We have expanded the references in the revised manuscript.

  1. Why is the reference VDD/2 used in Fig. 2? How did the authors guarantee that this reference will establish the maximum current flowing in the reference inverter? 

REPLY: The inverter stages have been carefully designed (by properly adjusting the width of PMOS and NMOS transistors) for a logic threshold equal to VDD/2 in nominal conditions. Furthermore the proposed body bias loop in Fig. 2 guarantees a static output voltage of the inverter stages equal to VDD/2, regardless of the bias current set to tune the delay, thus guaranteeing a balanced input output voltage transfer characteristic and therefore maximum current for Vin=VDD/2.

  1. Can the authors comment on the operation of the opamp with 0.5V supply? How can you guarantee that all transistors are in the saturation region? 

REPLY:Referring to the OTA in Fig. 3, the minimum supply voltage to avoid that any transistor enters the triode region is set by the following inequalities:

-VDD/2-VGS1>VDS,sat8

-VDD-|VGS3|-(VDD/2-VGS1) >VDS,sat1

From the above equations it is evident that the OTA cannot be properly biased with the MOS transistors operating in strong inversion. However, by exploiting subthreshold operation, gate source voltages in the range between 150mV and 200mV are feasible, and amplifier can be properly biased without any transistor in triode.

  1. The output frequency deviation of the oscillator with the presence of process/mismatch should be elaborated. 

REPLY: we thank the reviewer for this comment. Mismatch Monte Carlo simulations have been performed. In typical conditions the mean value and standard deviation of the oscillation frequency have been found to be 516 MHz and 8.1 MHz respectively. The histogram of mismatch Monte Carlo simulations has been added in section 4 of the revised manuscript.

  1. The line sensitivity of the oscillator should be discussed. With the 0.5V supply voltage, I suspect that the oscillator is very sensitive to voltage variations. 

REPLY: On this point we do not agree with the reviewer. In fact the proposed body bias feedback loop forces the static output voltage to VDD/2 in spite of VDD variations and at the same time sets the bias current according to the specified reference current again in spite of VDD variations. This point has been better clarified in the revised manuscript. Since the current is set by the loop the proposed RO is much more reliable than conventional ring oscillators based on simple inverter gates.

To quantify the line sensitivity of the proposed RO, Table 2, 3 and 4 have been added in the revised manuscript. These tables summarize the corner analysis results of the main parameters of the simulated 5-stage current-controlled ring oscillator (nominal IBIAS equal to 350 nA and at 30 °C) for three values of the supply voltage VDD.

  1. How did the oscillation frequency behave amid process variations? 

REPLY: Variations with process corners are summarized in tables 2, 3 and 4 of the revised manuscript. Furthermore, mismatch Monte Carlo simulations of the oscillation frequency have been added in Fig. 14 of the revised manuscript.

Reviewer 2 Report

This paper presents a ring oscillator controlled by current and biased by bulk. The paper is well structured, and it is well analyzed. The proposed biasing strategy and implementation are interesting, and results are well reported.

On the other hand, I find a small problem around the 10fF in the load capacitance in the inverted. I find this capacitance very small, and it is in the range of the capacitance that can be implemented but as this capacitance is small my questions is:

can the layout implementation disturb the results? Could you do a layout of your circuit and make a simulation with the post-layout extracted circuit and present the results? These results can be very interesting to validate the circuit or to change the bias current to control the frequency.

Reviewer 3 Report

Interesting work but some additional clarification and data are necessary to prove the circuit idea will work robustly.

1) Please include phase noise simulation with respect to the temperature, and voltage, and process variation. Preferably it can be expressed using some form of eye diagrams. 

2) Wider range temperature (0 deg C to 60 deg C for example) are necessary for the design to be useful. Currently only standard temperature 30C is presented. 

3) In Table 3, please state the FOM formula used in your manuscript rather than just citing [16] which is a book. There are so many pages in the book and it is impossible for readers to gauge the exact formula used. 

4) It is not clear whether the results are post-layout simulation with the parasitics extracted? This is especially important for sub-threshold designs. Please state clearly. The results will not mean much if this is schematic level simulation without parasitics included.

5) Please show the layout of your design with the chip area dimension. Higlighting area used by each sub-circuit block.

6) Specifically how the body voltage is applied to each transistors? Obviously you need a triple well process, assuming Deep-N well NMOS are used, how much area penalty is involved compared to regular NMOS? Might be good idea to share your layout technique to readers.

7) Is body voltage confined to VDD/2? What if VDD fluactuates? Is there a compensating range of body voltage with respect to different operating voltage and temperature? 

Reviewer 4 Report

This paper proposes a current-controlled CMOS ring oscillator  which uses the bulk voltages of  inverter stages  to tune the oscillation frequency. The proposal is interesting and well evaluated. I suggest to accept the manuscript. I just propose the authors to extend the application scenario in IoT by including references for systems that can benefit from their proposal such as temperature sensors (e.g. Zambrano et al, "A 0.05 mm2, 350 mV, 14 nW Fully-Integrated Temperature Sensor in 180-nm CMOS", TCAS-II 2021) and more general sensors (e.g. Meng et al., "A Low-Power Relaxation Oscillator With Switched-Capacitor Frequency-Locked Loop for Wireless Sensor Node Applications", JSSCL 2019).

Author Response

Comments and Suggestions for Authors

This paper proposes a current-controlled CMOS ring oscillator  which uses the bulk voltages of  inverter stages  to tune the oscillation frequency. The proposal is interesting and well evaluated. I suggest to accept the manuscript. I just propose the authors to extend the application scenario in IoT by including references for systems that can benefit from their proposal such as temperature sensors (e.g. Zambrano et al, "A 0.05 mm2, 350 mV, 14 nW Fully-Integrated Temperature Sensor in 180-nm CMOS", TCAS-II 2021) and more general sensors (e.g. Meng et al., "A Low-Power Relaxation Oscillator With Switched-Capacitor Frequency-Locked Loop for Wireless Sensor Node Applications", JSSCL 2019).

REPLY: We thank the reviewer for this comment. We have added the suggested interesting references in the revised manuscript.

Round 2

Reviewer 1 Report

The author have addressed my comments.

Reviewer 3 Report

I have no further comments.